# Metabolic Syndrome Including Glycated Hemoglobin A1c in Adults: Is It Time to Change?

**DOI:** 10.3390/jcm8122090

**Published:** 2019-12-01

**Authors:** Iván Cavero-Redondo, Vicente Martínez-Vizcaíno, Celia Álvarez-Bueno, Cristina Agudo-Conde, Cristina Lugones-Sánchez, Luis García-Ortiz

**Affiliations:** 1Universidad de Castilla-La Mancha, Health and Social Research Center, 16071 Cuenca, Spain; ivan.cavero@uclm.es (I.C.-R.); vicente.martinez@uclm.es (V.M.-V.); 2Universidad Politécnica y Artísitica del Paraguay, 001518 Asunción, Paraguay; 3Universidad Autónoma de Chile, Facultad de Ciencias de la Salud, 3460000 Talca, Chile; 4Institute of Biomedical Research of Salamanca (IBSAL), Primary Care Research Unit of Salamanca (APISAL), Health Service of Castilla y León (SACyL), 37003 Salamanca, Spain; cagudoconde@yahoo.es (C.A.-C.); crislugsa@gmail.com (C.L.-S.); Lgarciao@usal.es (L.G.-O.); 5Department of Biomedical and Diagnostic Sciences, University of Salamanca, 37003 Salamanca, Spain

**Keywords:** metabolic syndrome, vascular health, HbA1c

## Abstract

(1) Background: To assess the suitability of replacing conventional markers used for insulin resistance and dysglycemia by HbA1c in both the quantitative and qualitative metabolic syndrome (MetS) definition criteria; (2) Methods: Confirmatory factorial analysis was used to compare three quantitative definitions of MetS that consisted of many single-factor models, one of which included HbA1c as the dysglycemia indicator. After that, the model with the better goodness-of-fit was selected. Furthermore, a new MetS qualitative definition was proposed by replacing fasting plasma glucose with HbA1c > 5.7% in the International Diabetes Federation (IDF) definition. The clinical performance of these two MetS criteria (IDF and IDF-modified including HbA1c as the dysglycemia indicator) to predict vascular damage (pulse wave velocity [PWv], intima media thickness [IMT] and albumin-to-creatinine ratio [ACR]) was estimated; (3) Results: The single-factor model including HbA1c showed the better goodness-of-fit (χ^2^ = 2.45, df = 2, *p* = 0.293, CFI = 0.999, SRMR = 0.010). Additionally, the IDF-modified criteria gained in clinical performance to predict vascular damage (diagnostic Odds Ratio: 6.94, 1.34 and 1.90) for pulse wave velocity (PWv), intima media thickness (IMT) and albumin-to-creatinine ratio (ACR), respectively; and (4) Conclusions: These data suggest that HbA1c could be considered as a useful component to be included in the MetS definition.

## 1. Introduction

As conceptualized by several expert committees, metabolic syndrome (MetS) is a cluster of cardiometabolic risk factors that includes central obesity, dyslipidemia, insulin resistance, dysglycemia and elevated blood pressure [1,2]. Previous evidence has clearly demonstrated that the presence of MetS almost doubles the risk of cardiovascular disease mortality, myocardial infarction and stroke, and increases the risk of all-cause mortality by 1.6 times [3]. However, a criticism for the usefulness of the MetS diagnosis is that it fails to predict cardiovascular risk better than the sum of individual components [4]. Moreover, counting risk factors is not sensitive to reflect little changes in risk factors.

Therefore, from clinical and research settings it has been suggested that there is a need to include continuous variables in the definition of MetS [5]. The use of a single indicator MetS score calculated from the continuous key variables has some advantages, particularly for research. Dichotomizing the MetS key variables reduces the statistical power, especially for variables in which there is not a clear threshold for categorizing cardiometabolic risk, such as blood pressure [6]. Several MetS scores have been proposed during the last two decades [7,8,9,10], but none of them, or any of the MetS definition criteria, have included glycated hemoglobin A1c (HbA1c) as the dysglycemia indicator.

All this despite the growing evidence demonstrating the association between HbA1c and cardiovascular events and mortality [11], as well as with the biomarkers of vascular and renal damage, such as the pulse wave velocity (PWv), the intima-media thickness (IMT) and the albumin-to-creatinine ratio (ACR) [12,13,14]. Furthermore, HbA1c has been recognized by the American Diabetes Association (ADA) [15] and the World Health Organization (WHO) [16] as a biomarker for the diagnosis and control of diabetes mellitus type 2, since it has been demonstrated to have a better diagnostic accuracy for diabetes retinopathy than fasting plasma glucose (FPG) [17].

Thus, the aim of this study was to assess the suitability of replacing conventional markers used for insulin resistance and dysglycemia (FPG, the homeostasis assessment model for insulin resistance [HOMA-IR] or fasting insulin) with HbA1c in both quantitative and qualitative MetS criteria.

## 2. Experimental Section

This was a cross-sectional analysis of 1243 healthy adults, aged 18–91 years, from the EVIDENT II baseline data (trial registration number: NCT02016014), whose study protocol has been published elsewhere [18]. The participants were selected through random sampling from general practitioners’ offices belonging to six primary care centres from many cities in Spain. Anthropometry, blood pressure and biochemical determinations were performed using standard procedures. This study was approved by the Research Ethics Committee of the Salamanca University Hospital and all participants provided written informed consent according to the general recommendations of the Declaration of Helsinki [19].

### 2.1. Variables

For biochemical determinations, a blood sample was taken from the cubital vein between 08:00 and 09:00 after at least 12 h of fasting, and abstaining from smoking, alcohol and caffeinated beverages. FPG, high-density lipoprotein cholesterol (cHDL), creatinine and triglyceride concentrations were measured using standard enzymatic automated methods, and HbA1c and urine albumin were measured using an immune-turbidimetric assay. The fasting insulin was determined using a chemiluminescent microparticle immunoassay. The insulin sensitivity was determined using the HOMA-IR index with the following formula: fasting plasma glucose (mmol/l) × fasting insulin (mU/mL)/22.5. The ACR was estimated as urine albumin (mg/dL)/creatinine (mg/L).

The waist circumference was measured using a flexible graduated measuring tape with the patient in the standing position without clothing. The upper border of the iliac crests was located, and the tape was wrapped above this point, parallel to the floor, ensuring that it was adjusted without compressing the skin.

The blood pressure was calculated as the mean of the last two out of three measurements of the systolic blood pressure (SBP) and the diastolic blood pressure (DBP) using a validated OMRON model M10-IT sphygmomanometer (Omron Health Care, Kyoto, Japan), according to the recommendations of the European Society of Hypertension [20]. The mean arterial pressure (MAP) was estimated as SBP + 1/3 (SBP-DBP).

The arterial stiffness measurement was performed during the morning, following the blood samples. The PWv was estimated using the SphymgoCor System (AtCor Medical Pty Ltd. Head Office, West Ryde, Sydney, Australia). The PWv was measured with the patient in the supine position, estimating the delay in pulse wave at the carotid and femoral level as compared to the electrocardiogram wave [21]. One trained investigator performed the PWv measurements.

The IMT measurements were performed using a Sonosite Micromax ultrasound device (SonoSite Ltd., Herts, UK) paired with a 5–10 MHz multi-frequency high-resolution linear transducer with Sonocal software. A carotid ultrasound to assess the IMT was performed by two trained investigators. The common carotid was measured after the examination of a 10 mm longitudinal section at a distance of 1 cm from the bifurcation, performing measurements in the proximal and in the distal wall in the lateral, anterior and posterior projections, following an axis perpendicular to the artery to discriminate two lines, one for the intima-blood interface and the other for the media-adventitious interface. A total of six measurements were obtained of the right carotid and another six of the left carotid, and the average values of these measurements were considered for the analysis of the IMT [22].

### 2.2. Statistical Analysis

#### 2.2.1. MetS as Quantitative Scale

In order to examine the construct validity of a single-factor model for measuring MetS, three models were examined: (A) proposed by Pladevall et al. [9], including the waist circumference, the triglyceride-to-cHDL ratio, the HOMA-IR and the MAP; (B) proposed by Martinez-Vizcaino et al. [23], including the waist circumference, the triglyceride-to-cHDL ratio, the fasting insulin and the MAP; and (C) a new model that included HbA1c rather than HOMA-IR or fasting insulin as the dysglycemia indicator. Factor loadings of >0.3 and a statistical significance of (*p* < 0.05) were considered as criteria to be included in the MetS construct [24]. Because the χ^2^ test in studies with a large sample size tends to show a significant lack of fit in the models tested, in the present study a model was deemed to have a good fit when the comparative fit index (CFI) was >0.96 and the root mean square residual (SRMR) was <0.08 [25]. Additionally, the single-factor models were estimated for comparing the goodness-of-fit by the following subgroups: sex (women and men); weight status (normal weight and overweight/obesity); cHDL level (normal-cHDL [≤40 mg/dL] and low-cHDL [>40 mg/dL]); triglycerides level (normal-triglycerides [≤150 mg/dL] and high triglycerides [>150 mg/dL]); waist circumference (normal waist circumference [≤88 cm women/≤102 cm men] and increased waist circumference [>88 cm women/>102 cm men]); hypertension status (yes or no); diabetes mellitus status (yes or no); and pharmacological treatment status (yes or no).

A summative MetS index of each model was calculated by adding the standardized scores for the variables included in each model, multiplied by their weight factors.

Finally, to examine the relationship of each proposed summative index with the vascular health parameters (PWv, IMT and ACR) the Pearson correlation coefficients were calculated.

#### 2.2.2. MetS as a Dichotomic Scale

A new MetS criteria (IDF-modified) was proposed by replacing FPG with HbA1c in the International Diabetes Federation (IDF) definition [26], using HbA1c ≥ 5.7% as a prediabetes cut off [15]. The mean differences in the vascular health parameters between those with or without MetS following the IDF criteria or the IDF-modified criteria were compared using the Student-*t* test.

Finally, the performances of these two MetS criteria to predict vascular damage according to the PWv (≥10 m/s) [21], IMT (≥0.8 mm) [27] and ACR (≥30 mg/g) [28] were assessed. The sensitivity, specificity, positive likelihood ratio (PLR), negative likelihood ratio (NLR), diagnostic odds ratio (dOR) (which could take values from 0 to infinity; a value of 1 indicates null diagnostic ability of the test, while higher values show better diagnostic performance) [29] and accuracy (proportion of subjects correctly classified) [30] were calculated.

The criteria for statistical significance were set to 0.05. All statistical analyses were performed using Stata SE software, version 15 (StataCorp, College Station, TX, USA).

## 3. Results

The age of participants ranged from 18 to 91 (mean 55.94 ± 13.69) years old. Of these, 739 (59.5%) were women. The prevalence of MetS according to the IDF criteria was 49.2% (Table 1).

### 3.1. MetS as a Quantitative Scale

Figure 1 depicts the three single-factor models proposed for the analysis of the factorial structure of MetS. Because the factor loading for insulin in Model B (λ = 0.07) was lower than 0.3, this model was considered to not meet the minimal validity requirements. A better goodness-of-fit was observed in Model C, which included HbA1c (χ^2^ = 2.45, df = 2, *p* = 0.293, CFI = 0.999, SRMR = 0.010), even though the factor loading for HOMA-IR (λ = 0.62) in Model A was greater than for HbA1c (λ = 0.41) in Model C. Additionally, Model C showed a good fit for all subgroups (sex, weight status, cHDL level, triglycerides level, waist circumference, hypertension status diabetes mellitus status and pharmacological treatment status). Conversely, Model A showed less-of-goodness-fit when the single-factor model for measuring MetS was performed in women (CFI = 0.958), participants with an increased waist circumference (CFI = 0.949), hypertensive participants (CFI = 0.958) and non-diabetic participants (CFI = 0.955). Finally, Model B showed less goodness-of-fit in women (CFI = 0.948), patients who were overweight/obese (CFI = 0.946), had normal-cHDL (CFI = 0.950), an increased waist circumference (CFI = 0.945), non-diabetic participants (CFI = 0.940) and participants on pharmacological treatment (CFI = 0.955).

Furthermore, the correlation coefficients among the three summative indexes of each single-factor model were associated with the vascular health parameters. Model C (HbA1c) showed the highest correlation coefficient values 0.56 (*p* < 0.001) for PWv, 0.33 (*p* < 0.001) for IMT and 0.09 (*p* < 0.01) for ACR (Figure 2).

### 3.2. MetS as a Quantitative Scale

The differences in PWv between participants with or without MetS (Table 2), depending on the MetS criteria used (IDF or IDF-modified), were statistically significant (*p* < 0.001). However, these differences were statistically significant for the IMT (*p* = 0.028) and the ACR (*p* = 0.015) only when the participants were classified using the IDF-modified criteria.

Considering the dOR as a global statistic of the performance of a diagnostic test (30), the IDF-modified criteria classified the participants’ vascular damage better (Table 3): 6.94 (95%CI: 3.02–15.95), 1.34 (95%CI:0.61–2.93) and 1.90 (CI: 1.16–3.10) for PWv, IMT and ACR, respectively. Overall, the IDF-modified criteria performed better than the traditional IDF criteria on other accuracy statistics.

## 4. Discussion

In recent years, HbA1c has proven to be a biomarker for both cardiovascular and metabolic risk; however, none of the scientific consensus for the diagnosis of MetS has included HbA1c as a component factor of the MetS criteria. Our data, using confirmatory factor analysis, demonstrate that a single-factor model including HbA1c shows better construct validity than two previously validated models that differ in their dysglycemic markers: HOMA-R [9] and fasting insulin [23]. Moreover, this study proposes new MetS definition criteria that replace FPG with HbA1c as the dysglycemic marker in the traditional IDF criteria, which improves the accuracy for correctly classifying patients according to their vascular damage.

Although insulin resistance was placed for a long time in the cluster of risk factors for cardiovascular events and diabetes that make up MetS, the pathogenic mechanisms that support this hypothesis are far from being clarified. Thus, although neither clinicians nor researchers question the usefulness of HOMA-IR, the insulin-based syndrome hypothesis has been replaced by new hypotheses that pose the excess of energy intake as the nucleus of the pathogenetic mechanisms underlying MetS [2]. In addition, because insulin assays are not available in all clinical laboratories [31,32], in the latest consensus to define MetS FPG was considered as the biochemical indicator to represent dysglycemia [1,33]. Because HbA1c reflects the long-term glycemic control in diabetic patients [34], and since our data supported that the goodness-of-fit of the single-factor model that included this biochemical parameter was better than other models previously validated, we proposed a quantitative MetS index that includes HbA1c as a validated measure to be used in the clinical setting, but more importantly in research ones. The correlation coefficients with the PWv, IMT and ACR, which were slightly higher than those other models, support this proposal.

Our findings show that the correlational differences between the two valid models, A (HOMA-IR) and C (HbA1c), were not substantive. Since HOMA-IR and HbA1c measure different characteristics of dysglycemia, these correlational differences could be due to two different etiologies of MetS. Furthermore, these two different etiologies could be two different stages of dysglycemia: (i) in an early stage, in which there is only an increase in the insulin secreted by the pancreas to achieve the acceptable blood glucose levels, there are high insulin levels but slightly elevated or even normal blood glucose levels; and (ii) in a second stage, in which the pancreatic failure is beginning to appear, both blood glucose and HbA1c levels increase [35]. Additionally, insulin resistance has been associated with the activation of inflammatory cell adhesion molecules and cytokines, while prolonged hyperglycemia has been associated with an increase in oxidative stress [36] (a mechanism associated with ACR [37]).

Recent studies have shown HbA1c levels between 5.45% and 5.65% to be predictors of MetS in non-diabetic subjects [38,39]. Our study examined the suitability of the IDF-modified criteria that include, according with the prediabetes ADA criterion [15], an HbA1 level of 5.7% as a cut-off criterion, which would replace FPG as a dysglycemia indicator. According to our results, this change implies a considerable increase in the accuracy for predicting vascular damage associated with an increased cardiometabolic risk, as suggested by the greater differences in the PWv, IMT and ACR means between those with and without MetS classified according to these IDF-modified criteria. Likewise, the diagnostic performance of these new criteria is better than that of the original IDF (Table 3).

Improvements in both the quantitative and qualitative definitions of MetS agree with previous research, indicating that HbA1c is a good predictor of mortality and cardiovascular events [11]. Additionally, these findings are in line with previous evidence of a strong association between HbA1c and skin autofluorescence-indicated advanced glycation end-products (AGEs) [40], a tissue biomarker considered an indicator closely related with PWv, IMT and ACR [41], and associated with mortality and cardiovascular events [42].

The results of this research demonstrate that it could be feasible, and easily implementable into the clinical practice routine, to include the HbA1c in the MetS IDF criteria. Despite these characteristics, it is not to be expected that changes in the MetS definition criteria will be enthusiastically received by clinicians and policymakers, since the challenges of filling the gap between the publication of evidence and its implementation into practice have been consistently reported [43,44]. Our research claims to address this gap, notwithstanding that the single-factor model including HbA1c is useful in research and could potentially be implemented in clinical practice through mHealth strategies such as developing a mobile-based MetS calculator app, similar to existing ones [45].

Our results should be cautiously interpreted since they come from a cross-sectional study and, therefore, do not establish a temporal relationship between the MetS indexes and cardiovascular or metabolic events, in such a way that we cannot test its predictive validity. Also, the MetS indexes have been calculated using a Spanish specific population and, unless the sociodemographic characteristics and cardiovascular risk profile are similar to those of our study population, they cannot be compared with other cardiometabolic indexes used to date. In this study, we included only the measures that are most frequently associated with MetS. Recent studies have expanded the MetS concept to include other physiological variables such as uric acid, inflammation, procoagulation and vitamin K-dependent protein [46,47]. Future studies should investigate the influence of these elements on the factorial structure of the syndrome.

## 5. Conclusions

Our findings confirm that a single-factor model underlies the MetS concept and support that a quantitative MetS index that includes HbA1c as the dysglycemic marker is more closely associated to vascular health parameters such as the PWv, IMT and ACR. Also, this study demonstrates that the IDF-modified criteria that include HbA1c rather than FPG as the dysglycemic marker have better accuracy in classifying patients according to their vascular damage, thus supposedly according to their atherosclerotic-related cardiovascular and metabolic risk. Because the prevalence of MetS is rising rapidly due to the elevated prevalence of sedentary behaviours, the obesity pandemic and aging populations, improving the accuracy of its definition and its early diagnosis by introducing HbA1c, which would not substantially increase measurement costs, could represent an affordable approach.

## Figures and Tables

**Figure 1 jcm-08-02090-f001:**
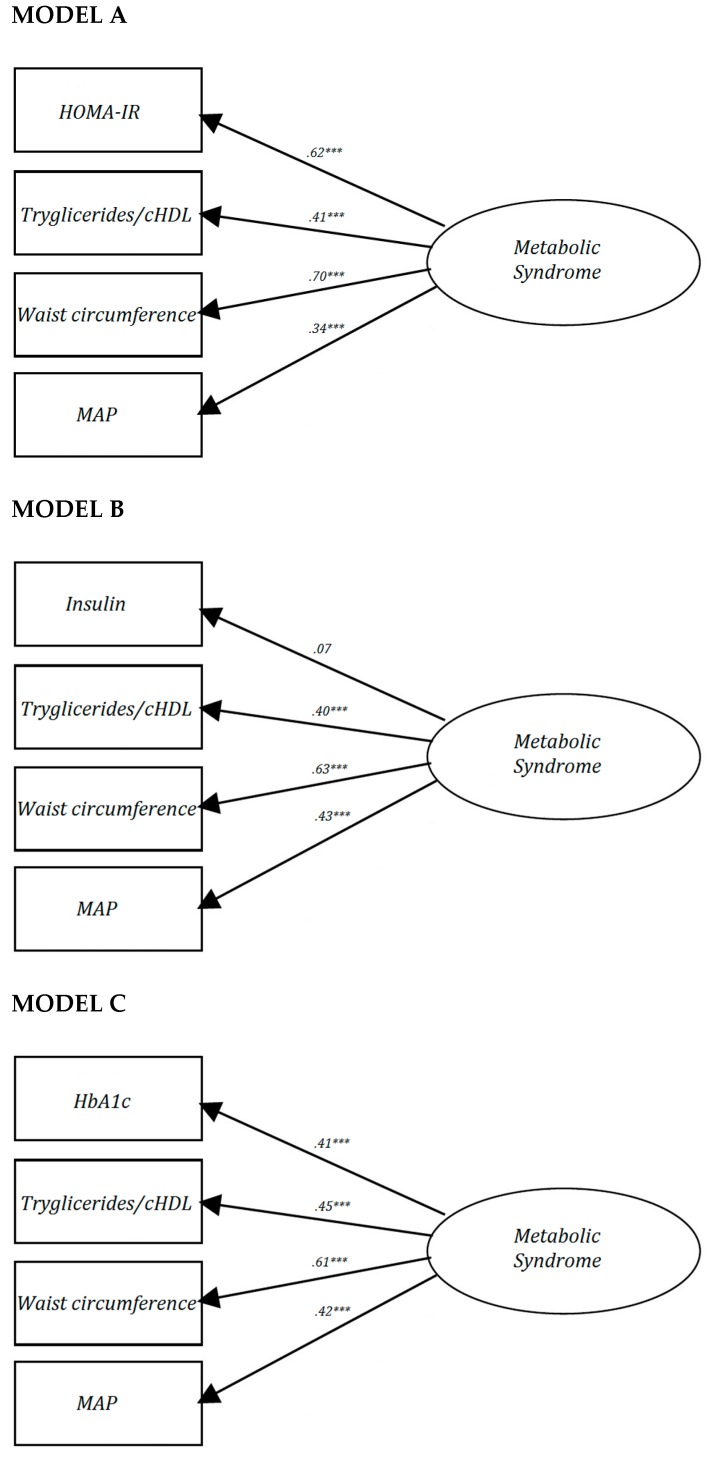
Factor loading and goodness-of-fit indexes of single-factor models for metabolic syndrome. *** *p* < 0.001. HOMA-IR: homeostasis assessment model for insulin resistance; cHDL: high density lipoprotein cholesterol; MAP: mean arterial pressure; χ^2^: chi-square; df: degree freedom; CFI: comparative fit index; SRMR: root mean square residual; HbA1c: glycated hemoglobin A1c. MODEL A: proposed by Pladevall et al. [9]; MODEL B: proposed by Martinez-Vizcaino et al. [23]; and MODEL C: a new model that included HbA1c rather than HOMA-IR or fasting insulin as the dysglycemia indicator.

**Figure 2 jcm-08-02090-f002:**
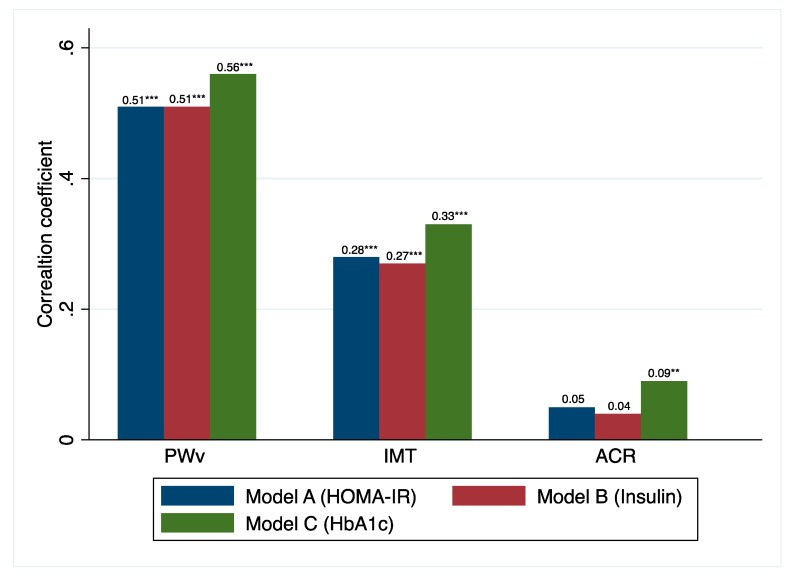
Correlation coefficients between each metabolic syndrome index model and PWv, IMT and ACR. *** *p* < 0.001, ** *p* < 0.01. PWv: pulse wave velocity; IMT: intima media thickness; ACR: albumin-to-creatinine ratio; HOMA-IR: homeostasis assessment model for insulin resistance; HbA1c: glycated hemoglobin A1c.

**Table 1 jcm-08-02090-t001:** Characteristics of the EVIDENT II population included in this analysis.

Variables	Total *n* = 1243	Men *n* = 504	Women *n* = 739
MetS prevalence (%) ^1^	611 (49.2)	262 (52.0)	349 (47.2)
Age (years)	55.94 ± 13.69	56.48 ± 13.65	54.11 ± 13.65
Waist circumference (cm)	93.40 ± 12.60	98.68 ± 11.22	89.82 ± 12.24
SBP (mmHg)	125.06 ± 16.94	129.70 ± 14.83	122.31 ± 17.78
DBP (mmHg)	77.43 ± 10.43	79.04 ± 10.45	75.65 ± 11.03
MAP (mmHg)	51.43 ± 13.97	55.80 ± 13.33	48.46 ± 13.63
HbA1c (%) (*n* = 1231)	5.65 ± 0.75	5.73 ± 0.80	5.60 ± 0.71
FPG (mg/dL)	93.81 ± 23.10	97.32 ± 25.81	91.44 ± 20.76
Fasting insulin (µIU/mL) (*n* = 1153)	8.09 ± 6.09	8.39 ± 6.16	7.91 ± 6.03
HOMA-IR (*n* = 1150)	1.93 ± 1.70	2.07 ± 1.76	1.84 ± 1.66
cHDL (mg/dL)	58.83 ± 15.58	51.94 ± 12.47	63.51 ± 15.74
Tryglicerides (mg/dL)	123.54 ± 118.47	143.36 ± 143.36	110.08 ± 95.89
PWv (m/s) (*n* = 243)	7.65 ± 2.01	8.24 ± 2.29	7.23 ± 1.68
IMT (mm) (*n* =247)	0.68 ± 0.11	0.71 ± 1.20	0.66 ± 0.09
ACR (mg/g) (*n* = 1041)	10.99 ± 36.43	10.44 ± 28.96	11.38 ± 40.85

Values are presented in mean ± SD and number (%). ^1^ Following International Diabetes Federation (IDF) criteria. MetS: metabolic syndrome; SBP: systolic blood pressure; DBP: diastolic blood pressure; MAP: mean arterial pressure; HbA1c: glycated hemoglobin A1c; FPG: fasting plasma glucose; HOMA-IR: homeostasis assessment model for insulin resistance; cHDL: high density lipoprotein cholesterol; PWv: pulse wave velocity; IMT: intima media thickness; ACR: albumin-to-creatinine ratio.

**Table 2 jcm-08-02090-t002:** Mean differences in PWv, IMT and ACR between subjects without and with metabolic syndrome.

	Without MetS Mean ± SD	MetS Mean ± SD	Mean difference	*p*
**IDF criteria**				
PWv (m/s)	7.10 ± 1.67 (*n* = 133)	8.33 ± 2.20 (*n* = 109)	−1.23	<0.001
IMT (mm)	0.67 ± 0.11 (*n* = 134)	0.70 ± 0.10 (*n* = 112)	−0.03	0.053
ACR (mg/g)	9.37 ± 34.10 (*n* = 520)	12.69 ± 38.76 (*n* = 516)	−3.32	0.143
**IDF with HbA1c criteria**				
PWv (m/s)	7.17 ± 1.68 (*n* = 178)	8.96 ± 2.01 (*n* = 65)	−1.80	<0.001
IMT (mm)	0.67 ± 0.11 (*n* = 180)	0.71 ± 0.10 (*n* = 67)	−0.03	0.028
ACR (mg/g)	9.27 ± 33.29 (*n* = 746)	15.36 ± 43.12 (*n* = 295)	−6.09	0.015

MetS: Metabolic syndrome; IDF: International Diabetes Federation; PWv: pulse wave velocity; IMT: intima media thickness; ACR: Albumin-to-creatinine ratio; HbA1c: glycated hemoglobin A1c.

**Table 3 jcm-08-02090-t003:** Accuracy parameters in the discrimination of vascular/renal risk parameters, by metabolic syndrome criteria.

	*n*	Sensitivity (%)	Specificity (%)	PLR	NLR	dOR	Accuracy ^a^
**PWv (>10 m/s)**							
IDF criteria	242	79.3 (52.7–100.0)	59.6 (50.1–71.0)	1.96 (0.16–24.63)	0.35 (0.02–5.11)	5.66 (2.21–14.48)	0.62 (0.14–2.76)
IDF with HbA1c criteria	243	65.5 (41.8–100)	78.5 (67.5–91.3)	3.05 (0.23–40.48)	0.44 (0.04–5.32)	6.94 (3.02–15.95)	0.77 (0.14–4.20)
**IMT (>0.8mm)**							
IDF criteria	246	50.0 (31.1–80.4)	54.9 (45.8–65.8)	1.11 (0.05–23.86)	0.91 (0.04–18.02)	1.22 (0.59–2.51)	0.54 (0.13–2.20)
IDF with HbA1c criteria	247	32.4 (17.9–58.4)	73.7 (63.0–86.2)	1.23 (0.03–44.56)	0.92 (0.06–15.01)	1.34 (0.61–2.93)	0.68 (0.13–3.37)
**ACR (>30mg/g)**							
IDF criteria	1036	56.9 (41.9–77.3)	50.7 (46.4–55.4)	1.16 (0.06–21.84)	0.85 (0.04–17.70)	1.36 (0.84–2.21)	0.51 (0.13–2.02)
IDF with HbA1c criteria	1041	41.7 (29.1–59.6)	72.7 (67.5–91.3)	1.52 (0.06–37.55)	0.80 (0.05–12.48)	1.90 (1.16–3.10)	0.71 (0.14–3.62)

^a^ The accuracy was defined as the number of correctly classified participants/number of all participants. PLR: positive likelihood ratio; NLR: negative likelihood ratio; dOR: diagnostic odds ratio; IDF: International Diabetes Federation; PWv: pulse wave velocity; IMT: intima media thickness; ACR: Albumin-to-creatinine ratio; HbA1c: glycated hemoglobin A1c.

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
