# Peer review of "Metabolic Syndrome Including Glycated Hemoglobin A1c in Adults: Is It Time to Change?"

_jcm, 2019, doi:10.3390/jcm8122090_

Round 1

Reviewer 1 Report

-Consider re-wording the sentence from 45-47. It is hard to understand as currently worded.

-were patients with your sample set on medication or was this an exclusion. How would your model deal with patients on medications for given criteria? Current, qualitative yes/no criteria, is able to account for medication treatment of these specific factors.

-The correlational differences between the two valid models, A and C, are not huge (and all are signficant with the exception of ACR) however, as you correctly discuss the use of HOMA-IR versus HbA1C% measures very different features of dysglycemia. Is it possible that two different etiologies of MetS exist whereby one model may be better at predicting the risk factors for a certain population versus the other?

-Can you discuss the actual clinical implementation of your proposed model. An appealing feature of qualitative models for clinicians is the ease at which one can determine MetS (a certain number of yes/no). Is there a need for the development of software to assist in the implementation of such quantitative models?

-

Author Response

Specific comment

-Consider re-wording the sentence from 45-47. It is hard to understand as currently worded.

Authors

Thanks for the reviewer’s comment. As suggested, we have re-worded the mentioned sentence as follows:

“Therefore, from clinical and research settings it has been suggested the needed to include continuous variables in the definition of MetS”

Specific comment

-were patients with your sample set on medication or was this an exclusion. How would your model deal with patients on medications for given criteria? Current, qualitative yes/no criteria is able to account for medication treatment of these specific factors.

Authors

We really appreciate the thoughtful reviewer’s comment. Thus, we have compared the goodness-of-fit of single-factor models by pharmacological treatment status, and included this information in the Statistical Analysis and Results sections as follows:

In the Statistical Analysis section:

“[…]Additionally, single-factor models were estimated for comparing the goodness-of-fit by the following subgroups: sex (women and men); weight status (normal weight and overweight/obesity); cHDL level (normal-cHDL [≤40mg/dL] and low-cHDL [>40mg/dL]); triglycerides level (normal-triglycerides [≤150mg/dL] and high triglycerides [>150mg/dL]); waist circumference (normal waist circumference [≤88 cm women/≤102 cm men] and increased waist circumference [>88 cm women/>102 cm men]); hypertension status (yes or no); diabetes mellitus status (yes or no); and pharmacological treatment status (yes or no).”

In the Results section:

“[…] Additionally, Model C showed a good fit for all subgroups (sex, weight status, cHDL level, triglycerides level, waist circumference, hypertension status diabetes mellitus status and pharmacological treatment status). Conversely, Model A showed less goodness fit when single-factor model for measuring MetS were performed in women (CFI = 0.958), increased waist circumference (CFI = 0.949), hypertensive participants (CFI = 0.958) and non-diabetic participants (CFI = 0.955). Finally, Model B showed less goodness fit in women (CFI = 0.948), overweight/obesity (CFI = 0.946), normal-cHDL (CFI = 0.950), increased waist circumference (CFI = 0.945), non-diabetic participants (CFI = 0.940) and participants on pharmacological treatment (CFI = 0.955).”

Specific comment

-The correlational differences between the two valid models, A and C, are not huge (and all are signficant with the exception of ACR) however, as you correctly discuss the use of HOMA-IR versus HbA1C% measures very different features of dysglycemia. Is it possible that two different etiologies of MetS exist whereby one model may be better at predicting the risk factors for a certain population versus the other?

Authors

The reviewer´s comment seems judicious. As suggested, we have included in the discussion section a new paragraph considering this issue. Additionally, we have included three new references.

In the Discussion section

“Our findings show that the correlational differences between the two valid models, A (HOMA-IR) and C (HbA1c), were not substantive. Since HOMA-IR and HbA1c measure different characteristics of dysglycemia, these correlational differences could be due to two different etiologies of MetS. Furthermore, these two different etiologies could be two different stages of dysglycemia: i) in an early stage, in which there is only an increase in the insulin secretion by the pancreas to achieve the acceptable blood glucose levels, there are high insulin levels but slightly elevated or even normal blood glucose levels; and ii) in a second stage, in which the pancreatic failure is beginning to appear, both blood glucose  and  HbA1c levels increase[36]. Additionally, insulin resistance has been associated with the activation of inflammatory cell adhesion molecules and cytokines, while prolonged hyperglycemia has been associated with an increase in oxidative stress [37](mechanism associated with ACR [38]).”

In References:

American Diabetes Association. Diagnosis and classification of diabetes mellitus. Diabetes care 2013; 36(Supplement 1): S67-S74. Lee Y, Fluckey JD, Chakraborty S, Muthuchamy M. Hyperglycemia and hyperinsulinemia-induced insulin resistance causes alterations in cellular bioenergetics and activation of inflammatory signaling in lymphatic muscle. The FASEB Journal 2017; 31(7): 2744-2759 Mels CM, Huisman HW, Smith W, Schutte R, Schwedhelm E, Atzler D, Böger RH, Ware LJ, Schutte AE. The relationship of nitric oxide synthesis capacity, oxidative stress, and albumin-to-creatinine ratio in black and white men: the SABPA study. Age 2016; 38(1): 9.

Specific comment

-Can you discuss the actual clinical implementation of your proposed model. An appealing feature of qualitative models for clinicians is the ease at which one can determine MetS (a certain number of yes/no). Is there a need for the development of software to assist in the implementation of such quantitative models?

Authors

Thank you for reviewer´s comment. As suggested, we have included in the discussion section a paragraph about clinical implementation.

In the Discussion section

“The results of this research demonstrate that could be feasible, and easily implementable into the clinical practice routine, to include the HbA1c in the MetS IDF criteria. Despite these characteristics, it is not to be expected that changes in the MetS definition criteria will be enthusiastically received by clinicians and policymakers, since the challenges for filling the gap between the publication of evidence and its implementation into practice have been consistently reported (44, 45). Our research claims to address this gap, notwithstanding that the single-factor model including HbA1c is useful in research and could potentially be implemented in clinical practice through mHealth strategies such as developing a mobile-based MetS calculator app, similar to existing ones (i.e. http://www.movidavida.org/calcumovi).”

In References:

Graham ID, Logan J, Harrison MB, Straus SE, Tetroe J, Caswell W, Robinson N. Lost in knowledge translation: time for a map?. Journal of continuing education in the health professions 2006; 26(1): 13-24. Lang ES, Wyer PC, Haynes RB. Knowledge translation: closing the evidence-to-practice gap. Annals of emergency medicine 2007; 49(3): 355-363.

Reviewer 2 Report

This is an interesting study aimed at looking for a criterion of dyglycemia which would improve the association between the metabolic syndrome and marlers of vascular damage (PWV, IMT and ACR). The Authors demonstrated benefits of introducing HbA1c into the definition of the metabolic syndrome and were able to show advantages of their approach in comparison with traditional definitions of the syndrome.

I have one comment: they are advised to compare the superiority of their approach across various subroups of the study subjects: e.g. women and men; normal-weight, overweight and obese; normal and low-HDL; normal and high TG; normal and increased waist circumference; normotensive and hypertensive subjects; normoglycemic and hyperglycemic patients.

Author Response

This is an interesting study aimed at looking for a criterion of dysglycemia which would improve the association between the metabolic syndrome and markers of vascular damage (PWV, IMT and ACR). The Authors demonstrated benefits of introducing HbA1c into the definition of the metabolic syndrome and were able to show advantages of their approach in comparison with traditional definitions of the syndrome.

Authors

We would like to thank the kind reviewer´s comment and the time spent on the revision of this paper.

Specific comment

I have one comment: they are advised to compare the superiority of their approach across various subgroups of the study subjects: e.g. women and men; normal-weight, overweight and obese; normal and low-HDL; normal and high TG; normal and increased waist circumference; normotensive and hypertensive subjects; normoglycemic and hyperglycemic patients.

Authors

We really appreciate the thoughtful reviewer’s comment. As suggested, we have performed single factor models to compare the goodness-of-fit by the reviewer´s proposed subgroups. We have included this information in the Statistical Analysis and Results sections as follows:

In the Statistical Analysis section:

“[…]Additionally, single-factor models were estimated for comparing the goodness-of-fit by the following subgroups: sex (women and men); weight status (normal weight and overweight/obesity); cHDL level (normal-cHDL [≤40mg/dL] and low-cHDL [>40mg/dL]); triglycerides level (normal-triglycerides [≤150mg/dL] and high triglycerides [>150mg/dL]); waist circumference (normal waist circumference [≤88 cm women/≤102 cm men] and increased waist circumference [>88 cm women/>102 cm men]); hypertension status (yes or no); diabetes mellitus status (yes or no); and pharmacological treatment status (yes or no).”

In the Results section:

“[…] Additionally, Model C showed a good fit for all subgroups (sex, weight status, cHDL level, triglycerides level, waist circumference, hypertension status diabetes mellitus status and pharmacological treatment status). Conversely, Model A showed less goodness fit when single-factor model for measuring MetS were performed in women (CFI = 0.958), increased waist circumference (CFI = 0.949), hypertensive participants (CFI = 0.958) and non-diabetic participants (CFI = 0.955). Finally, Model B showed less goodness fit in women (CFI = 0.948), overweight/obesity (CFI = 0.946), normal-cHDL (CFI = 0.950), increased waist circumference (CFI = 0.945), non-diabetic participants (CFI = 0.940) and participants on pharmacological treatment (CFI = 0.955).”